# Study of Prescription-Indication of Outpatient Systemic Anti-Fungals in a Colombian Population. A Cross-Sectional Study

**DOI:** 10.3390/antibiotics11121805

**Published:** 2022-12-13

**Authors:** Luis Fernando Valladales-Restrepo, Juan Alberto Ospina-Cano, Brayan Stiven Aristizábal-Carmona, Diana Fiorella López-Caicedo, Melissa Toro-Londoño, Andrés Gaviria-Mendoza, Manuel Enrique Machado-Duque, Jorge Enrique Machado-Alba

**Affiliations:** 1Grupo de Investigación en Farmacoepidemiología y Farmacovigilancia, Universidad Tecnológica de Pereira-Audifarma S.A, Pereira 660001, Colombia; 2Grupo de Investigación Biomedicina, Facultad de Medicina, Fundación Universitaria Autónoma de las Américas, Pereira 660001, Colombia; 3Semillero de Investigación en Farmacología Geriátrica, Facultad de Medicina, Fundación Universitaria Autónoma de las Américas, Pereira 660001, Colombia

**Keywords:** antifungal agents, ketoconazole, drug prescriptions, inappropriate prescribing, pharmacoepidemiology, Colombia

## Abstract

The inappropriate use of antifungals is associated with greater antimicrobial resistance, costs, adverse events, and worse clinical outcomes. The aim of this study was to determine prescription patterns and approved and unapproved indications for systemic antifungals in a group of patients in Colombia. This was a cross-sectional study on indications for the use of systemic antifungals in outpatients from a drug dispensing database of approximately 9.2 million people affiliated with the Colombian Health System. Sociodemographic, pharmacological, and clinical variables were considered. Descriptive, bivariate, and multivariate analyses were performed. A total of 74,603 patients with antifungal prescriptions were identified; they had a median age of 36.0 years (interquartile range: 22.0–53.0 years), and 67.3% of patients were women. Fluconazole (66.5%) was the most prescribed antifungal for indications such as vaginitis, vulvitis, and vulvovaginitis (35.0%). A total of 29.3% of the prescriptions were used in unapproved indications. A total of 96.3% of ketoconazole users used the medication in unapproved indications. Men (OR: 1.91; CI95%: 1.79–2.04), <18 years of age (OR: 1.20; CI95%: 1.11–1.31), from the Caribbean region (OR: 1.26; CI95%: 1.18–1.34), with chronic obstructive pulmonary disease (OR: 1.80; CI95%: 1.27–2.54), prescriptions made by a general practitioner (OR: 1.17; CI95%: 1.04–1.31), receiving comedications (OR: 1.58; CI95%: 1.48–1.69), and the concomitant use of other antimicrobials (OR: 1.77; CI95%: 1.66–1.88) were associated with a higher probability that the antifungal was used for unapproved indications; deep mycosis (OR: 0.49; CI95%: 0.41–0.58), prescribing fluconazole (OR: 0.06; CI95%: 0.06–0.06), and having diabetes mellitus (OR: 0.33; CI95%: 0.29–0.37), cancer (OR: 0.13; CI95%: 0.11–0.16), or HIV (OR: 0.07; CI95%: 0.04–0.09) reduced this risk. Systemic antifungals were mostly used for the management of superficial mycoses, especially at the gynecological level. In addition, more than a quarter of patients received these medications in unapproved indications, and there was broad inappropriate use of ketoconazole.

## 1. Introduction

Antimicrobial drug resistance is a major health concern worldwide. It has been classified by the World Health Organization (WHO) as one of the main global threats to public health [1]. It is estimated that almost 1 billion people suffer from fungal infections of the skin, nails, and hair, and more than 150 million people have serious fungal diseases, which have a significant impact on their lives and are sometimes fatal [2]. The prevalence of drug-resistant fungal infections is increasing and worsens an already difficult therapeutic situation [1]. This leads to difficulties in the treatment of these infections, therapeutic failures, longer hospital stays, and higher cost options [1]. The factors that lead to antifungal resistance are diverse and heterogeneous, highlighting the increasing human exposure to antifungals, as well as the use of these antimicrobials in farm animals and crops, in addition to the changes in the use of agricultural chemicals, inadequate waste management, and climate change, among others [3].

The excessive use and misuse of antimicrobials have been considered one of the main causes of drug resistance; as a result, emphasis has been placed on appropriately prescribed medications [4,5,6,7]. However, minimum conditions to achieve an adequate formulation of antifungals and antivirals need to be determined [6,7]. There are few published studies on the patterns of use of systemic antifungals in the outpatient setting [8,9]. In addition, no studies on the use of antifungals could be found to determine their appropriate or inappropriate use in this group of patients. Studies have addressed this topic but involve only hospitalized patients, reflecting that the inappropriate use of these drugs is 25.0% to 80.2% [10,11,12,13,14,15,16].

The Colombian Health System offers universal coverage to the entire population through two affiliation regimes: the contributory one that is paid by workers and employers and the subsidized one that is responsible for the insurance of all people without the ability to pay and includes a benefit plan that involves a significant number of systemic antifungals. The objective of this study was to determine the prescription patterns of systemic antifungals and their use in approved and unapproved uses in a group of outpatients in Colombia in 2022.

## 2. Results

A total of 74,603 patients in 192 cities were identified who had a first prescription for a systemic antifungal. A total of 67.3% (*n* = 50,224) were women, and the median age was 36.0 years (interquartile range: 22.0–53.0 years; range: 0.0–104.0 years). A total of 16.5% (*n* = 12,322) were younger than 18 years (<12 years: *n* = 7539; 10.1%), 34.8% (*n* = 25,945) were 18–39 years, 30.7% (*n* = 22,897) were 40–64 years, and 10.5% (*n* = 7870) were 65 or older. A total of 7.5% (*n* = 5569) had no recorded age.

According to geographic regions, patients were mainly found in the Caribbean (*n* = 43,814; 58.7%), followed by Bogotá-Cundinamarca (*n* = 12,442; 16.7%), Central (*n* = 8932; 12.0%), Pacific (*n* = 6205; 8.3%), and Eastern-Amazonia-Orinoquía (*n* = 3210; 4.3%). A total of 55.4% (*n* = 41,323) contributory health insurance and 44.6% (*n* = 33,280) had subsidized health insurance.

A total of 30.3% (*n* = 22,629) of patients had some chronic pathology. The five most common comorbidities were arterial hypertension (*n* = 11,424; 15.3%), diabetes mellitus (*n* = 3541; 4.7%), hypothyroidism (*n* = 2271; 3.0%), dyslipidemia (*n* = 1754; 2.4%), and cancer (*n* = 1506; 2.0%).

### 2.1. Systemic Antifungals

The most prescribed antifungal was fluconazole (*n* = 49,591; 66.5%), followed by nystatin (*n* = 13,257; 17.8%), and ketoconazole (*n* = 9748; 13.1%) (Table 1). A total of 1.0% (*n* = 753) used two or more systemic antifungals, especially the combination of fluconazole with ketoconazole (*n* = 335/753; 44.5%) and fluconazole with nystatin (*n* = 333/753; 44. 2%). Of the ketoconazole users, 97.3% (*n* = 9485/9748) had not previously received any other antifungal. Tablets or capsules were the most frequent pharmaceutical forms (*n* = 61,855; 82.9%), followed by oral suspensions (*n* = 13,108; 17.6%) and oral solutions (*n* = 39; 0.1%). This group of medications was prescribed mainly by general medical practitioners (*n* = 70,270; 94.2%), followed by specialists (*n* = 1753; 2.3%), surgeons (*n* = 1436; 1.9%), and dentists (*n* = 1144; 1.5%).

A primary or secondary diagnosis related to fungal infections was identified in 48.0% (*n* = 35,824) of patients. Superficial infections occurred in 30.6% (*n* = 22,841) of them, while deep infections were identified in 1.6% (*n* = 1175) of the cases. In 6.4% (*n* = 4773) of the cases, mycosis could not be classified, and in 9.4% (*n* = 7035) of the patients, the infection was not directly related to a fungal etiology. A total of 70.7% (*n* = 25,310/35,824) of the antifungals were prescribed for approved uses, especially for vaginitis, vulvitis, and vulvovaginitis (*n* = 12,555; 35.0%), and 29.3% (10,514/35,824) of the medications were prescribed for unapproved indications, mainly in patients with acute rhinopharyngitis (*n* = 1002; 2.8%). Table 2 shows the main approved and unapproved uses.

Ketoconazole was the antifungal that was used in greater proportion for off-label purposes (*n* = 3802/3949; 96.3%), followed by nystatin (*n* = 3336/5272; 63.3%), terbinafine (*n* = 145/796; 18.2%), fluconazole (*n* = 3307/25,340; 13.1%), itraconazole + secnidazole (*n* = 6/48; 12.5%), posaconazole (*n* = 2/18; 11.1%), itraconazole (*n* = 64/588; 10.9%), voriconazole (*n* = 4/38; 10.5%). and fluconazole + secnidazole (*n* = 9/102; 8.8%). Ketoconazole was used mainly for the management of vaginitis–vulvitis–vulvovaginitis and for tinea versicolor (see Appendix A).

### 2.2. Comedications

A total of 28.1% (*n* = 20,988) of patients received concomitant antimicrobials, especially systemic antibiotics (*n* = 11,590; 15.5%), antiprotozoal (*n* = 6984; 9.4%), anthelminthic (*n* = 4190; 5.6%), or antivirals (*n* = 625; 0.8%). A total of 48.6% (*n* = 36,264) of all patients also received topical antifungals cutaneously (*n* = 31,611; 42.4%) or vaginally (*n* = 12,488; 16.7%). The most commonly used comedication in the three months prior to the index date was with analgesics and anti-inflammatories (*n* = 32,671; 43.8%), followed by micronutrients and nutritional supplements (*n* = 20,540; 27.5%), anti-ulcer medications (*n* = 14,649; 19.6%), antihistamines (*n* = 14,147; 19.0%), and antihypertensives (*n* = 11,601; 15.6%).

### 2.3. Multivariate Analysis

The binary logistic regression found that men, patients under 18 years of age, individuals from the Caribbean region, prescriptions made by general practitioners, patients with chronic obstructive pulmonary disease, those receiving comedications, and the concomitant use of other antimicrobials were associated with a higher probability that the antifungal was used off-label. Having a diagnosis of deep mycosis, a prescription for fluconazole, and comorbidities such as diabetes mellitus, cancer, or HIV reduced this risk (Table 3).

## 3. Discussion

This study allowed us to characterize the prescription pattern of systemic antifungals and their approved and unapproved uses as evidence of drug use among patients affiliated with the Colombian Health System. These findings can be useful for health care, academic, and scientific personnel in making decisions regarding the risks faced by their patients and contribute to strengthening the practices regarding appropriate use of antifungals among physicians as a way to reduce antimicrobial resistance.

Fluconazole was the most prescribed antifungal in this group of patients, which was consistent with findings in the United States by Benedict et al. (75.0%) [8] and reported in Australia by Wang et al. (73.5%) [9]. Similarly, these studies of outpatients were consistent with what was found in studies of hospitalized patients (40.0–80.5%) [14,15,16,17,18,19]. When comparing our results with the data of the European Surveillance of Antimicrobial Consumption Network (ESAC-Net) on antifungals prescribed for outpatients in 2020 in 27 countries of the European Union and two countries of the European Economic Area, it was found that, on average, the DHDs (defined daily dose per 1000 inhabitants per day) of terbinafine, voriconazole, and itraconazole were lower, while those of fluconazole and ketoconazole were higher than those found in Europe [20]. Variations in the pattern of drug use are multifactorial and may depend on the preferences of the prescriber, adherence to management guidelines, availability of the drug in the health systems of each country, resistance patterns, and sensitivity of the drug microorganisms in each region, among others [21,22,23].

Systemic antifungals were used in approved uses mainly for vulvovaginal infections. In the United States, the majority of women with vulvovaginitis due to *Candida* sp. were treated with systemic fluconazole (70.0%) [24]. In Germany, the majority were treated with topical antifungals such as clotrimazole (72.0%) [25]. According to clinical practice guidelines, the treatment of vulvovaginitis due to *Candida* sp. depends on its classification as uncomplicated (immunocompetent patients with infrequent episodes and mild symptoms) or complicated (≥4 episodes/year, with severe symptoms or in immunosuppressed patients) [26,27,28]. Uncomplicated cases can be managed with local or systemic antifungals such as fluconazole, while complicated cases can be managed with systemic antifungals [26,27,28].

This study found that 29.3% of medications were prescribed in unapproved indications. There are few published studies that address this topic, with only a few on the inappropriate use of systemic antifungals in the hospital environment [10,11,12,13,14,15,16]. For example, in Greece, researchers found that 25.0% of antifungal prescriptions were inappropriate [10], similar to that described in Spain (25.0%) [11] and Oman (25.2%) [12]. In France, inappropriate use was higher (30.0–40.0%) [13,14], while in Brazil nonadherence to management guidelines was even more worrisome (64.6–80.2%) [15,16]. No studies on the indication prescription of antifungal drugs were found in Colombia, but previously it was reported that 23.5 to 31.3% of patients who received outpatient antibiotics had them prescribed for unapproved uses [21,22,23]. The inappropriate use of these drugs contributes to the increasing microbial resistance, which can cause adverse events and increase hospitalization costs and mortality rates [4,5]. Therefore, the optimal selection of the antifungal, dose, route of administration, and duration of therapy are key to preserving the efficacy of antimicrobials [4].

The antifungal that was used in the greatest proportion for unapproved uses was ketoconazole. According to the ESAC-Net, in 20 European countries, there were no outpatient dispensations of ketoconazole in 2020 [9]. In Australia, from 2005 to 2013, ketoconazole was prescribed for 40.5–56.1% of patients, and after 2013, there were no dispensations [9]. In the United States in 2018, the medication was used in only 1.0% of patients who received systemic antifungals [8]. This contrasts markedly with what was found in this study. The FDA (2013) announced that oral ketoconazole should not be used as first-line treatment of fungal infections due to the risk of hepatotoxicity, adrenal insufficiency, and potential contraindicated or major drug interactions [29], leading to the medication’s withdrawal in many countries [30]. However, ketoconazole continues to be used in Colombia [31] but carries a health alert from INVIMA (2014), which states that oral pharmaceutical forms of ketoconazole are indicated only in cases in which there are no other therapeutic options or when these are not tolerated in patients with fungal infections that put life at risk [32]. We believe that the high use of ketoconazole in Colombia is due to a marked lack of adherence to clinical practice guidelines and ignorance of the health alerts issued regarding its toxicity. Although the cost of ketoconazole is low and could be another reason for its high prescription, in the country there is access to fluconazole, which is equally as cheap and has the advantage of being safer [31]. It is important to promote continuing education programs and pharmacovigilance strategies to improve patient safety. Thus, a study carried out in Colombia showed that after a medical intervention, there was a reduction in the prescription of ketoconazole in 31.1% of cases [33].

Some variables related to presenting prescriptions in unapproved indications were found. For example, there was a higher probability of unapproved uses of prescriptions made by general medical practitioners, which was consistent with what was found in Greece in hospital formulations of antifungals (primary care physicians vs. specialists in infections; 35% vs. 5% *p* < 0.001, respectively) [10]. Similarly, men and younger individuals were associated with a higher probability of unapproved uses of medication, consistent with a study on the prescription of fluoroquinolones [23]. Other pharmacoepidemiological studies in Colombia had shown that inadequate prescriptions were more prevalent in certain regions of the country, evidencing the wide heterogeneity in the clinical behaviors of physicians [21,22,23], which was also found in this research. Patients with chronic obstructive pulmonary disease had a higher risk of receiving nonindicated antifungals, which was consistent with what was found in a study on the use of macrolides [21]. The exacerbations of these patients’ conditions were mainly caused by viruses, bacteria, and environmental factors [34]. On the other hand, patients who received other medications concomitantly were more likely to receive inadequate prescriptions, as identified in this research and in another local study [23]. The simultaneous use of different antimicrobials likely denoted the absence of clarity of diagnosis. Thus, in the USA, Filice et al. found that when a correct diagnosis was made, 62% of antimicrobial regimens were appropriate, compared to only 5% when the diagnosis was incorrect, undetermined, or when physicians treated a sign or symptom instead of a syndrome or disease (*p* < 0.001) [35].

Some limitations were recognized in the interpretation of the results, since access to the medical records were not obtained to verify the pathologies of the patients and thus confront the accuracy of the diagnoses assigned by the physician. In addition, it was only possible to identify the primary and secondary diagnoses associated with each prescription, where almost 50% corresponded to infections. The reports of the paraclinical studies that could have been carried out on the patients were not known. Similarly, the medications prescribed outside the health system or not delivered by the dispensing company were unknown. However, the health system had a significant number of subjects distributed in most of the national territory, involving both the contributory and subsidized health insurance systems.

## 4. Materials and Methods 

### 4.1. Study Design and Patients

A cross-sectional study was carried out to establish the prescriptions, patterns, and approved and unapproved uses of systemic antifungals in outpatients. A drug dispensing database that collects information from approximately 9.2 million people affiliated with the Colombian Health System was used. These patients subscribed to four health insurance companies, corresponding to approximately 30.0% of the active affiliated population of the contributory or payment regime and 6.0% of the state-subsidized regime, which comprised approximately 17.0% of the Colombian population [36].

Patients were identified from prescriptions for systemic antifungals from 1 April to 30 June 2022. The first prescription of the antifungal was considered the index date. Patients of any sex and age who were treated as outpatients were eligible. Patients with prescriptions of parenteral or topical pharmaceuticals were excluded. 

### 4.2. Variables

Based on the information about drug consumption in the affiliated population systematically obtained by the dispensing company (Audifarma SA) [36], a database was designed that allowed the following groups of patient variables to be collected:

Sociodemographics: age, sex, type of affiliation to the Colombian Health System (contributory or subsidized), and city of medication dispensation. The place of residence was categorized by departments according to the regions of Colombia, considering the classification of the National Administrative Department of Statistics-DANE of Colombia, as follows: Bogotá-Cundinamarca, Caribbean, Central, Eastern, Pacific, and Amazonia-Orinoquía regions.

Type of mycosis: the subjects’ illnesses were classified as superficial (skin, hair, nail, or mucosal infections), deep or systemic (paracoccidioidomycosis, histoplasmosis, coccidioidomycosis, aspergillosis, cryptococcosis, candidiasis, among others), and indeterminate (for diagnoses that could not be determined or classified), according to the codes of the international classification of diseases (ICD-10).

Chronic comorbidities: These conditions were identified from the main and secondary diagnoses reported 90 days prior to the index date using the ICD-10 codes.

Pharmacological: name of the prescribed antifungal (terbinafine, ketoconazole, fluconazole, itraconazole, voriconazole, posaconazole, isavuconazole, flucytosine, griseofulvin, and nystatin (the latter was included despite not having systemic effects, due to the similarity in the method of administration and the risk of unwanted gastrointestinal effects), pharmaceutical form (tablet, suspension, or oral solution) and dose. The defined daily dose (DDD) was used as the unit of measurement of drug use, according to WHO recommendations and expressed as DHD (defined daily dose per 1000 inhabitants per day) [37].

Type of prescriber: general practitioner, medical specialist (internal medicine, pediatrics, geriatrics, dermatology, etc.), surgeons (general, orthopedics, obstetrics, and gynecology, etc.), and dentistry.

Comedications: on the same index date, participants were grouped into the following medication categories: (a) systemic antibiotics, (b) antivirals, (c) antiprotozoa, (d) anthelminths, and (e) topical antifungals. In addition, in the 90 days prior to the index date medications in the following categories were also considered: (a) antidiabetics (oral and subcutaneous), (b) antihypertensives and diuretics, (c) lipid-lowering; (d) antiulcer drugs, (e) antidepressants, (f) anxiolytics and hypnotics (benzodiazepines and Z drugs), (g) thyroid hormone, (h) antipsychotics (typical and atypical), (i) antiepileptics, (j) analgesics, (k) bronchodilators, and (l) micronutrients and nutritional supplements, among others.

### 4.3. Rationality of Antifungals

Use of the medication was assessed considering primary and secondary diagnoses associated with each prescription according to the ICD-10 codes, and it was established whether the medication was approved or not approved for the diagnosis according to the Food and Drug Administration (FDA) of the United States [38] and the National Institute of Food and Drug Surveillance (INVIMA) of Colombia [31] (Appendix A). The use of ketoconazole without previous use of another systemic antifungal was considered an inadequate prescription [32].

### 4.4. Ethical Statement

The protocol was approved by the Bioethics Committee of the Technological University of Pereira in the category of research without risk (Endorsement code: 52-050922). The ethical principles established by the Declaration of Helsinki were respected.

### 4.5. Data Analysis

The data were analyzed with the statistical package SPSS Statistics, version 26.0 for Windows (IBM, Armonk, NY, USA). A descriptive analysis was performed with frequencies and proportions for the qualitative variables and measures of central tendency and dispersion for the quantitative variables, depending on their parametric behavior established by the Kolmogorov-Smirnov test. The comparison of categorical variables was performed using the *X*^2^ tests or Fisher’s exact test. A multivariate binary logistic regression model was developed that included the associated variables in the bivariate analyses, as well as those with sufficient plausibility or reported association to identify those that could be associated with the use of systemic antifungals in unapproved indications after adjustment. A level of statistical significance was determined at *p* < 0.05.

## 5. Conclusions

With these findings, we can conclude that systemic antifungals were mostly used for the management of superficial mycoses, especially at the gynecological level. In addition, more than a quarter of patients received these medications for unapproved indications, especially those using ketoconazole. These findings can be useful for clinicians who treat infections and for decision-makers to strengthen continuing education programs for prescribers.

## Figures and Tables

**Table 1 antibiotics-11-01805-t001:** Prescription patterns, frequency of use, dose, defined daily dose, and the distribution by sex and age in outpatients with systemic antifungal dispensations, Colombia.

Drug	*n* = 74.603	%	Dose (mg or IU/Day) ^1^				Sex		Age (Years)	
			Mean	Median	Mode	DHD	F (%)	M (%)	Mean (SD)	Median (IQR)
Fluconazole	49,591	66.5	1278.4	1000.0	800.0	0.38	72.0	28.0	44.6 ± 17.8	39.0 ± 27.0
Nystatin	13,257	17.8	8,646,752	6,000,000	6,000,000	0.09	54.8	45.2	25.5 ± 27.3	10.0 ± 46.0
Ketoconazole	9748	13.1	3854.2	4000.0	2000.0	0.23	61.1	38.9	36.9 ± 18.4	34.0 ± 29.0
Terbinafine	1421	1.9	6459.4	7000.0	7000.0	0.04	62.1	37.9	48.9 ± 20.2	52.0 ± 30.0
Itraconazole	1023	1.4	3191.1	2800.0	3000.0	0.02	63.3	36.7	45.1 ± 18.4	44.0 ± 29.0
Fluconazole + secnidazole	152	0.2	303.0	300.0	300.0	-	90.8	9.2	37.6 ± 13.3	34.0 ± 19.0
Itraconazole + secnidazole	80	0.1	654.7	399.6	399.6	-	90.0	10.0	36.0 ± 12.9	34.5 ± 16.5
Voriconazole	51	0.1	10,701.9	11,200.0	12,000.0	<0.01	35.3	64.7	37.1 ± 25.7	38.0 ± 47.5
Posaconazole	27	0.0	8933.3	8400.0	8400.0	<0.01	40.7	59.3	50.3 ± 19.4	50.0 ± 32.5
Isavuconazole	9	0.0	5988.9	5600.0	5600.0	<0.01	33.3	66.7	37.0 ± 9.0	37.0 ± 13.0

F: Feminine; M: Male; SD: Standard deviation; IQR: Interquartile range; DHD: Defined daily dose per 1000 inhabitants per day. ^1^ The unit of measurement of nystatin is in International Units (IU), while the rest of antifungals is in milligrams (mg).

**Table 2 antibiotics-11-01805-t002:** Approved and non-approved indications related to the dispensing of systemic antifungals in a group of patients in Colombia, 2022.

Diagnosis	Frequency *n* = 35.824	%
*Approved Indications*	25,310	70.7
Vaginitis–vulvitis–vulvovaginitis ^1^	12,555	35.0
Opportunistic mycoses	3226	9.0
Onychomycosis	1320	3.7
Superficial mycoses	1056	2.9
Unspecified mycosis	873	2.4
Body ringworm	850	2.4
Pityriasis versicolor ^2^	799	2.2
Urinary infection	790	2.2
Stomatitis	759	2.1
Unspecified dermatophytosis	564	1.6
Others (*n* = 43)	2518	7.0
*Unapproved Indications*	10,514	29.3
Acute nasopharyngitis	1002	2.8
Vaginitis–vulvitis–vulvovaginitis ^1^	842	2.4
Acute tonsillitis	785	2.2
Pityriasis versicolor ^2^	438	1.2
Bacterial or viral intestinal infections	415	1.2
Allergic/contact dermatitis	380	1.1
External otitis	339	0.9
Herpetic infections	312	0.9
Otitis media	283	0.8
Parasites	262	0.7
Others (*n* = 112)	5456	15.2

^1^ Fluconazole, griseofulvin, itraconazole, and nystatin are approved for the management of vulvovaginal candida infections, while the other antifungals are not (see Appendix A). ^2^ Fluconazole, itraconazole, and terbinafine are approved for the management of tinea versicolor, while the other antifungals are not (see Appendix A).

**Table 3 antibiotics-11-01805-t003:** Variables related to unapproved uses of systemic antifungals by binary logistic regression in 35,824 outpatients, Colombia, 2022.

Variables	*p*	OR	CI95%	
			Lower	Upper
Men	<0.001	1.917	1.797	2.045
Age <18 years	<0.001	1.208	1.111	1.312
Origin Caribbean region	<0.001	1.263	1.189	1.342
Prescription by general practitioner	0.009	1.171	1.040	1.319
Deep mycosis	<0.001	0.496	0.417	0.589
Fluconazole	<0.001	0.064	0.060	0.069
Diabetes mellitus	<0.001	0.331	0.291	0.376
Cancer	<0.001	0.138	0.117	0.162
Human Immunodeficiency Virus	<0.001	0.070	0.049	0.099
Chronic obstructive pulmonary disease	0.001	1.802	1.275	2.549
Receive systemic comedications	<0.001	1.586	1.483	1.696
Comedication with other antimicrobials	<0.001	1.774	1.665	1.889

OR: odds ratio; 95% CI: 95% confidence interval.

## Data Availability

(https://www.protocols.io/private/BE942CB1520A11EDB35B0A58A9FEAC02 to be removed before publication). Accesed: 22 October 2022.

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
