# Peer review of "Study of Prescription-Indication of Outpatient Systemic Anti-Fungals in a Colombian Population. A Cross-Sectional Study"

_antibiotics, 2022, doi:10.3390/antibiotics11121805_

Round 1

Reviewer 1 Report

The authors may discuss about causes and consequences of antifungal resistance.

Fisher MC, Alastruey-Izquierdo A, Berman J, Bicanic T, Bignell EM, Bowyer P, Bromley M, Brüggemann R, Garber G, Cornely OA, Gurr S. Tackling the emerging threat of antifungal resistance to human health. Nature Reviews Microbiology. 2022 Mar 29:1-5.

The full form of DHDs should be mentioned in the text on its first appearance.

Author Response

Manuscript ID: antibiotics-2019763
Type of manuscript: Article
Title: Study of prescription-indication of outpatient systemic antifungals in
a Colombian population

Dear Editors

Antibiotics

We respond point by point to the observations made by the reviewers.

Reviewer 1

The authors may discuss about causes and consequences of antifungal resistance.

Fisher MC, Alastruey-Izquierdo A, Berman J, Bicanic T, Bignell EM, Bowyer P, Bromley M, Brüggemann R, Garber G, Cornely OA, Gurr S. Tackling the emerging threat of antifungal resistance to human health. Nature Reviews Microbiology. 2022 Mar 29:1-5.

Answer: The reference is added and the causes that lead to antifungal resistance are added in the introduction.

The full form of DHDs should be mentioned in the text on its first appearance.

 Answer: Adjustment is done

Reviewer 2 Report

This manuscript, entitled “Study of prescription-indication of outpatient systemic antifungals in a Colombian population”, had identified a total of 74,603 patients with antifungal prescriptions. The authors classified various parameters of patients in Colombia, including sex, age, and geographic regions. And the authors also concluded the information of approved and non-approved indications and unapproved uses of systemic antifungals.

1.       First and most importantly, the authors analyzed the prescription patterns of systemic antifungals and their use in approved and unapproved used in 74,603 patients in Colombia in 2022 comprehensively. Have the authors received informed consent form from all patients?  

2.       Secondly, there is no information about the follow-up of patients after using the antifungals and the effect of the usage of antifungals. Did the authors have the information about the recovery situation of patients? If yes, please supplement the relevant information.

3.       Too limited information have been provided and analyzed in this study. What strains cause what diseases? What antifungals were used to treat what disease? In depth study and analysis should be performed to aid in better understanding. The current study is more like a news report than a scientific article.

4.       Overall, the manuscript is lack of organization and significance.

5.       In materials and methods section, please separate the information with subtitles. For example, “4.4 Statistical methods”. It will make this section more organized.

6.       Line spacing is not uniform. The line spacing between the last paragraph and the penultimate paragraph is apparently larger than others. Please check the whole manuscript and make them consistent.

7.       In results section, it would be better to provide serial number for “Systemic antifungals”, “Comedications”, “Multivariate analysis”, and keep each section of results has a subtitle.

Author Response

Manuscript ID: antibiotics-2019763
Type of manuscript: Article
Title: Study of prescription-indication of outpatient systemic antifungals in
a Colombian population

Dear Editors

Antibiotics

We respond point by point to the observations made by the reviewers.

Reviewer 2

This manuscript, entitled “Study of prescription-indication of outpatient systemic antifungals in a Colombian population”, had identified a total of 74,603 patients with antifungal prescriptions. The authors classified various parameters of patients in Colombia, including sex, age, and geographic regions. And the authors also concluded the information of approved and non-approved indications and unapproved uses of systemic antifungals.

First and most importantly, the authors analyzed the prescription patterns of systemic antifungals and their use in approved and unapproved used in 74,603 patients in Colombia in 2022 comprehensively. Have the authors received informed consent form from all patients?  

Answer: This is an observational cross-sectional and retrospective study. It has bioethics endorsement where the study was classified as risk-free research, therefore, according to Colombian regulations, informed consent is not required from each person. The confidentiality of the information was maintained.

Secondly, there is no information about the follow-up of patients after using the antifungals and the effect of the usage of antifungals. Did the authors have the information about the recovery situation of patients? If yes, please supplement the relevant information.

Answer: The study design did not involve follow-up of patients, and this information is not available. There is no information on these outcomes. This had already been reflected in the limitations of the study.

Too limited information have been provided and analyzed in this study. What strains cause what diseases? What antifungals were used to treat what disease? In depth study and analysis should be performed to aid in better understanding. The current study is more like a news report than a scientific article.

Answer: There is no microbiological information on cultures. It is added in limitations of the study. In addition, a supplementary table is added showing the top 15 uses according to the type of antifungal.

Overall, the manuscript is lack of organization and significance.

Answer: We try to organize it better, and accept the recommendations of all the reviewers to improve its content and order.

In materials and methods section, please separate the information with subtitles. For example, “4.4 Statistical methods”. It will make this section more organized.

Answer: Materials and methods are separated by subtitles

Line spacing is not uniform. The line spacing between the last paragraph and the penultimate paragraph is apparently larger than others. Please check the whole manuscript and make them consistent.

Answer: Adjustment is done

In results section, it would be better to provide serial number for “Systemic antifungals”, “Comedications”, “Multivariate analysis”, and keep each section of results has a subtitle.

Answer: Adjustment is done

Reviewer 3 Report

This manuscript describes the prescription patterns of systemic anti-fungals (various oral dosage forms) and their use in unapproved and approved indications in a large sample (n = 74603) of outpatients in Colombia. The manuscript is well-written for the most part and adheres to the journal’s guidelines. Yet, it can be improved using the below notes.

Better to indicate study design in the study title.

 Provide IQR along with the median value in the abstract.

 In abstract, it would be better to add odds ratio (95% CI) values next to all the predictors of irrational use of antifungal agents.

 Explain the reason of excluding prescriptions of topical antifungal medications and primarily focusing on the prescriptions containing oral dosage forms of antifungal agents.

 Better to divide Materials and Methods section into 4 or 5 appropriate subsections e.g. study design, database, study population (inclusion and exclusion criteria), ethical considerations, criteria for assessment of rationality of antifungals, data analysis etc.

Provide reference of the database used in the present study.

Section # 2, Paragraph 1: I’m not able to understand the age range (0.0-104.0 years). Was the minimum age 0? It would be good if the authors add information (number and percentage) of patients < 12 years old.

 Discussion section, paragraph # 4, authors need to elaborate all possible reasons of irrational use of ketoconazole. As physicians’ non-adherence to the standard treatment guidelines can be a key factor, it would be good to include data from the studies reporting physicians’ adherence to the treatment guidelines in Columbia and/or surrounding countries.

Author Response

Manuscript ID: antibiotics-2019763
Type of manuscript: Article
Title: Study of prescription-indication of outpatient systemic antifungals in
a Colombian population

Reviewer 3

This manuscript describes the prescription patterns of systemic anti-fungals (various oral dosage forms) and their use in unapproved and approved indications in a large sample (n = 74603) of outpatients in Colombia. The manuscript is well-written for the most part and adheres to the journal’s guidelines. Yet, it can be improved using the below notes.

Better to indicate study design in the study title.

Answer: Adjustment is done. “Study of prescription-indication of outpatient systemic anti-fungals in a Colombian population. A Cross-Sectional Study”

Provide IQR along with the median value in the abstract.

Answer: Adjustment is done.

In abstract, it would be better to add odds ratio (95% CI) values next to all the predictors of irrational use of antifungal agents.

Answer: Adjustment is done

Explain the reason of excluding prescriptions of topical antifungal medications and primarily focusing on the prescriptions containing oral dosage forms of antifungal agents.

Answer: Topical antifungals were excluded for the selection of the study population. We focus on the oral pharmaceutical forms because they are the ones that may be most related to safety problems. However, in the selected patients, those who concomitantly received topical pharmaceutical forms were identified. The above is embodied in the methodology.

Better to divide Materials and Methods section into 4 or 5 appropriate subsections e.g. study design, database, study population (inclusion and exclusion criteria), ethical considerations, criteria for assessment of rationality of antifungals, data analysis etc.

Answer: Adjustment is done

Provide reference of the database used in the present study.

Answer: The reference is added

Section # 2, Paragraph 1: I’m not able to understand the age range (0.0-104.0 years). Was the minimum age 0? It would be good if the authors add information (number and percentage) of patients < 12 years old.

Answer: It is right. The age range is from 0.0 to 104.0 years. The data is added:<12 years: n=7539; 10.1%.

Discussion section, paragraph # 4, authors need to elaborate all possible reasons of irrational use of ketoconazole. As physicians’ non-adherence to the standard treatment guidelines can be a key factor, it would be good to include data from the studies reporting physicians’ adherence to the treatment guidelines in Columbia and/or surrounding countries.

Answer: The discussion on this topic is expanded

Reviewer 4 Report

Nystatin carries significant systemic toxicity and is inappropriate for systemic use. Its efficacy is restricted to topical, oral and gastrointestinal infections. Thus, labelling it a ‘systemic antifungal’ is misleading; although it can be administered orally, its GI absorption is negligible.

Secnidazole is not an antifungal agent. It is categorized as an antiprotozoal and antibacterial drug. Thus, its inclusion in Table 1 has to be reconsidered or explained.

Explain why pityriasis versicolor appears under both approved and unapproved indications.

Likewise, vaginitis - vulvitis - vulvovaginitis also appears under both approved and unapproved indications.

Discuss why the prescription of ketoconazole is quite widespread in Colombia. Could low drug cost or limited availability of other agents (such as fluconazole) be the reason?

Author Response

Manuscript ID: antibiotics-2019763
Type of manuscript: Article
Title: Study of prescription-indication of outpatient systemic antifungals in
a Colombian population

Reviewer 4

Nystatin carries significant systemic toxicity and is inappropriate for systemic use. Its efficacy is restricted to topical, oral and gastrointestinal infections. Thus, labelling it a ‘systemic antifungal’ is misleading; although it can be administered orally, its GI absorption is negligible.

Answer: You are absolutely right, your observation is completely correct, however we are including it since it is administered orally and it can be associated with gastrointestinal side effects. We clarify it in the methods section.

Secnidazole is not an antifungal agent. It is categorized as an antiprotozoal and antibacterial drug. Thus, its inclusion in Table 1 has to be reconsidered or explained.

Answer: That is correct, but in the Colombian market there is a combination of fluconazole with seccnidazole and itraconazole with seccnidazole in the same medication. Being a drug for systemic use, it meets the inclusion criteria established in the study.

Explain why pityriasis versicolor appears under both approved and unapproved indications.

Answer: Fluconazole, itraconazole, and terbinafine are approved for the management of tinea versicolor while the other antifungals are not (see Supplementary Table 1). The clarification is made at the foot of table 2.

Likewise, vaginitis - vulvitis - vulvovaginitis also appears under both approved and unapproved indications.

Answer: Fluconazole, griseofulvin, itraconazole, and nystatin are approved for the management of vulvovaginal candida infections while the other antifungals are not (see Supplementary Table 1). The clarification is made at the foot of table 2.

Discuss why the prescription of ketoconazole is quite widespread in Colombia. Could low drug cost or limited availability of other agents (such as fluconazole) be the reason?

Answer: The discussion on this topic is expanded

Round 2

Reviewer 2 Report

This manuscript, entitled “Study of prescription-indication of outpatient systemic antifungals in a Colombian population”, had identified a total of 74,603 patients with antifungal prescriptions. The authors classified various parameters of patients in Colombia, including sex, age, and geographic regions. And the authors also concluded the information of approved and non-approved indications and unapproved uses of systemic antifungals. 

Comments:

1. The major concerns I mentioned in previous review were not addressed by the author. For example, this study is still lack of novelty and significance.

2. The paper has not been revised much and does not meet the criteria for publication in Antibiotics.

3. The data in this study are inadequate for publication.

Reviewer 4 Report

The authors have managed to adequately address all of my concerns. I have no further queries or comments.